# The protocol for a cluster randomized controlled trial to evaluate couple-based violence prevention education and its ability to reduce intimate partner violence during pregnancy in Southwest Ethiopia

Zeleke Dutamo Agde [1,2]*, Jeanette H. Magnus[3], Nega Assefa[4], Muluemebet Abera Wordofa[1]

1 Department of Population and Family Health, Institute of Health, Jimma University, Jimma, Ethiopia, 2 Department of Reproductive Health, College of Medicine and Health Sciences, Wachemo University, Hossana, Ethiopia, 3 Faculty of Medicine, University of Oslo, Oslo, Norway, 4 College of Health and Medical Sciences, Haramaya University, Harar, Ethiopia

* zeledutamo@gmail.com

**Data Availability Statement:** No datasets were generated or analysed during the current study. All

## Abstract

### Background

A significant proportion of women in Ethiopia suffer from violence by their intimate partner during pregnancy, which has adverse maternal and newborn outcomes. Couple-focused interventions are effective in reducing and/or controlling violence between women and their intimate partners. However, interventions addressing intimate partners of the victims are not well studied, particularly in the Ethiopian setting. This study aims to assess the effect of couple-based violence prevention education on intimate partner violence during pregnancy.

### Methods

We will use a cluster randomized controlled trial to evaluate the effectiveness of couple-based violence prevention education compared to routine care in reducing intimate partner violence during pregnancy. Sixteen kebeles will be randomly assigned to 8 interventions and 8 control groups. In the trial, 432 couples whose wife is pregnant will participate. Health extension workers (HEWs) will provide health education. Data will be collected at baseline and endline. All the collected data will be analyzed using Stata version 16.0 or SPSS version 25.0. We will use the McNemar test to assess the differences in outcomes of interest in both intervention and control groups before and after the intervention for categorical data. A paired t-test will be used to compare continuous outcome of interest in the intervention and the control groups after and before the intervention. The GEE (Generalized Estimating Equation), will be used to test the independent effect of the intervention on the outcome of the interest. Data analysis will be performed with an intention-to-treat analysis approach. During the analysis, the effect size, confidence interval, and p-value will be calculated. All tests will be two-sided, and statistical significance will be declared at p < 0.05.

relevant data from this study will be made available upon study completion.

**Funding:** The author(s) received no specific funding for this work.

**Competing interests:** The authors have declared that no competing interests exist.

## Discussion

We expect that the study will generate findings that can illuminate violence prevention strategies and practices in Ethiopia.

## Trial registration

It has been registered on ClinicalTrials.gov as NCT 05856214 on May 4, 2023.

## Background

In their lifetime, one in three women may experience physical and/or sexual abuse in an intimate relationship, with Africa and Southeast Asia having the highest rates [1,2]. Intimate partner violence during pregnancy is an important public health agenda and human rights violation [3–5].

Intimate Partner Violence (IPV) is a leading non-obstetric cause of maternal morbidity and mortality [6,7]. It carries a potential risk for both the mother and the unborn child. It leads to adverse maternal consequences such as antepartum hemorrhage, miscarriage, abortion, abruption placenta, preeclampsia, poor gestational weight gain and premature labor and maternal death [8–10]. Intimate partner violence during pregnancy is found to be associated with mental disorders like depression, anxiety, and sleep disorders [11,12]. Studies have shown that IPV during pregnancy is associated adverse new-born outcomes, including low birth weight, preterm birth, small for gestational age, and neonatal death [8,9,13]. The use of maternal health services is inversely correlated with intimate partner violence. Evidence from 36 national household surveys in low and middle-income countries by Leight and Wilson (2021) has found that IPV decreases the utilization of maternal health care (antenatal care, skilled delivery, and postnatal care) [14]. Moreover, recent evidence from a nationwide survey conducted in Ethiopia by Ousman et al. (2022) has shown that low maternal health care usage is associated with IPV during pregnancy [15].

A systematic review in Sub-Saharan Africa by Muluneh, M.D. et al. (2020) has shown that the prevalence of IPV is 44%, which is higher compared to other regions [16]. The prevalence of IPV during pregnancy in Ethiopia is one of the highest in the world, ranging from 26% to 65% [17–22].

Studies have shown that IPV is influenced by a multifaceted combination of individual, relational, community, and societal factors. These factors include younger age, lower education, substance abuse, rural living, unintended pregnancy, and individual-level attitudes towards IPV [19,22–25], power imbalance [26,27], polygamy [28], unhealthy communication [29], low social support, lower socioeconomic status [30], community gender inequitable norms and attitudes towards IPV [26], and public policies [31].

A recent systematic review and meta-analysis in low- and middle-income countries by Leight et al. (2023) has evidenced that group-based interventions have significantly reduced the likelihood of women experiencing IPV [32]. In sub-Saharan Africa, a number of interventions aimed at controlling and reducing IPV have been conducted [33–36], and these interventions have mostly been aimed at women. The engagement of men in interventions that target IPV prevention has been noted as crucial [37–39]. Couple-based interventions that have been aimed at preventing IPV have shown promise [40,41], but there is scarce evidence on effective and scalable interventions to reduce IPV during pregnancy that target couples in Ethiopia.

The trial finding is helpful to achieve Sustainable Development Goal (SDG) 5.2 calling for the elimination of all forms of violence against women and girls by the year 2030 [42]. It also

will provide a chance to advocate for the inclusion of the package in Ethiopia's Health Extension Package (HEP) and the Health Sectors Development Plan (HSDP) in Ethiopia. It is crucial to tailor interventions in a specific cultural context that involve both pregnant women and their husbands. Therefore, the aim of this trial is to examine the effect of Couple–Based Violence Prevention Education (CBVPE) on IPV during pregnancy in rural parts of southwest Ethiopia.

## Study objectives/hypothesis

### Research hypothesis

The proportion of IPV during pregnancy is lower among women who receive CBVPE than among couples who receive usual care from health extension workers.

### Primary objective

To evaluate the effect of CBVPE on intimate partner violence during pregnancy provided by health extension workers in the Hadiya Zone, Southwest Ethiopia

### Secondary objectives

- To examine the effect of CBVPE on couples' knowledge and attitudes, and controlling behavior toward intimate partner violence in the Hadiya Zone, Southwest Ethiopia.

- To examine the effect of CBVPE on women's autonomy and self-efficacy in the Hadiya Zone, Southwest Ethiopia

## Methods

### Trial design

A two arm parallel-group cluster randomized controlled trial with a 1:1 allocation ratio is designed to examine whether the CBVPE by HEWs reduces IPV during pregnancy in the Hadiya Zone, Southwest Ethiopia. Clusters will be kebeles found in selected districts of Hadiya zone, Hadiya,Ethiopia (Fig 1).

### Setting

The study will be conducted in the Hadiya Zone, which is one of the 11 administrative zones of Central Ethiopia. Its capital, Hossana Town, is located 235 kilometers southwest of the capital city, Addis Ababa. The Hadiya Zone has 13 woredas (districts) and 4 city adminstrations with an area of 3542.66 square kilometers and a population density of 444 people per square kilometer. The zone has 329 kebeles. According to the 2021 Hadiya Zone Statistics Office report, the zone had a total population of population of 1,767, 390 of these, 873,091(49.4%) are males and 894, 299 (50.6%) are females. Approximately 90% of the population lives in the rural part of the zone; Hadiya is the dominant ethnic group. Health care services are provided through 1 comprehensive specialized hospital, 3 primary hospitals, 61 health centers, and 311 health posts. In the Hadiya Zone, a total of 843 health extension workers are currently at work. The total number of women of childbearing age is 88,200.

For the trial, Soro, Lemo, Anlemo and Duna districts were randomly selected out of the 13 districts in the zone. The districts had a total population of 594,882, according to the 2021 Hadiya Zone Statistics report. Both urban and rural kebele (Ethiopia's lowest administrative

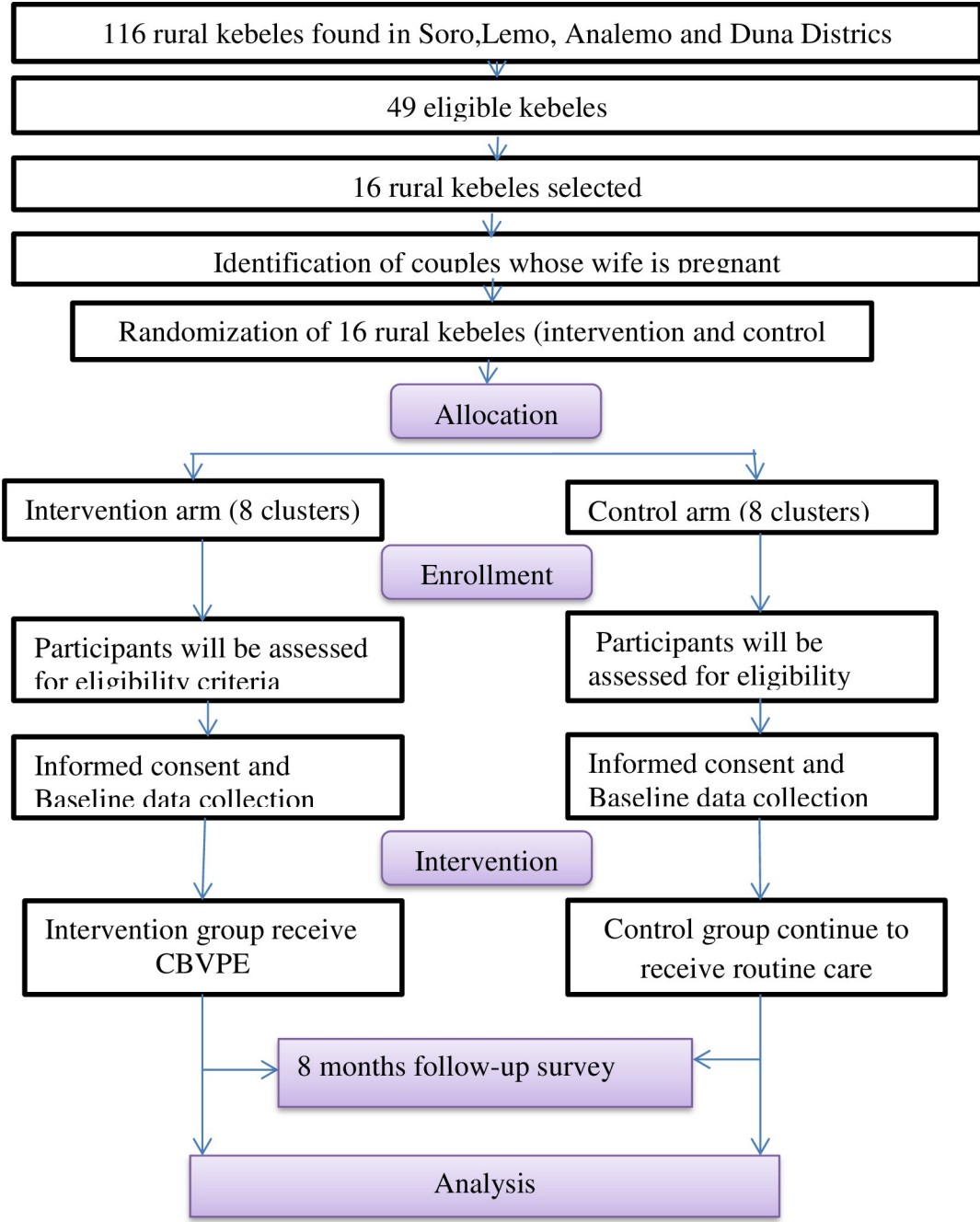

**Fig 1. Participant flow.**

unit) were found in the districts. There are 26, 33, 29, and 28 rural kebeles in Soro, Lemo, Anlemo and Duna districts, respectively. Health extension workers are key delivery agents for the Health Extension Program (HEP), which aims to ensure equitable community-level health care services [43]. Health extension workers are key delivery agents for the Health Extension Program (HEP), which aims to ensure equitable community-level health care services [32]. In each kebele, there is a health post that serves 3,000–5,000 villagers and is staffed by two HEWs [44]. The HEWs are selected from the community based on specific criteria, including being

female, aged 18+, having at least 10th grade education, and having the ability to speak the local language [45]. All HEWs received two years of training (one year of theoretical and one year of practical). Health extension workers spend 75% of their time on home-to-home visits and performing outreach activities in the community. They spend the remaining 25% of their time at the health post by providing services such as immunizations, injectable contraception, and basic curative services, including treating malaria, diarrheal disease, and intestinal parasites [46].

## Study population

A study will be conducted among pregnant women of childbearing age in selected intervention and control clusters.

## Eligibility criteria for clusters and HEWs

Out of 116 rural kebeles found in four districts (Soro, Lemo, Analemo, and Duna), 16 rural kebeles (8 intervention and 8 control groups) that are not adjacent and geographically accessible will be selected. Female HEWs, each from a selected kebele, who can speak the local language (Hadiyisa) will be selected as implementers of the intervention.

## Eligibility criteria for participants

The inclusion criteria for the participants in this trial will be a couple whose wife is pregnant, is in the first trimester, and has had at least one live birth. Couples who lived in the study area at least six months and whose wives' interpregnancy interval was less than two years are also inclusion criteria for the trial. The exclusion criteria for the participants in this trial will be either couples with severe psychological illness, which interferes with consent and trial participation, or severely ill or plan to move out of the intervention and control groups for the next 8 months.

## Sample size determination

The sample size is determined using Stata version 16.0, taking into consideration the assumed intra-cluster correlation coefficient, the effect size and study's power. We hypothesize that the CBVPE will reduce the proportion of IPV during pregnancy from 37.5% (based on the study report from Ofla district, Ethiopia [19]) to 17.5% in the intervention group. To detect a 20% difference in the rates of IPV, intra-cluster correlation coefficient of 0.05 [39], a power of 80% and a type I error (alpha) of 5% for a two-tailed test are used. The design effect of 2.3 is considered to account for the lack of independence between participants within clusters and to improve the power of the study. Design effect is defined as $1+\rho$ (m-1), where $\rho$ is the intra-cluster correlation coefficient and m is the number of couples per cluster, is set to be 27. By taking into account a 20% loss to follow-up, a total of 432 couples (216 couples in the intervention groups and 216 in the control groups) will be included in the study.

## Sampling procedures

Four districts (Soro, Lemo, Analemo, and Duna) are selected randomly from the 13 districts in the zone, which consists of 116 kebeles. Distance and geographical accessibility will be considered to select interventions and control kebeles. At least one kebele will be left between the kebele included in the trial to serve as a buffer zone between the two arms in order to avoid information contamination. Out of 116 rural kebeles, 49 non-adjacent and accessible kebeles are identified. Finally, Out of 49, 16 rural kebeles—the lowest administrative units from the

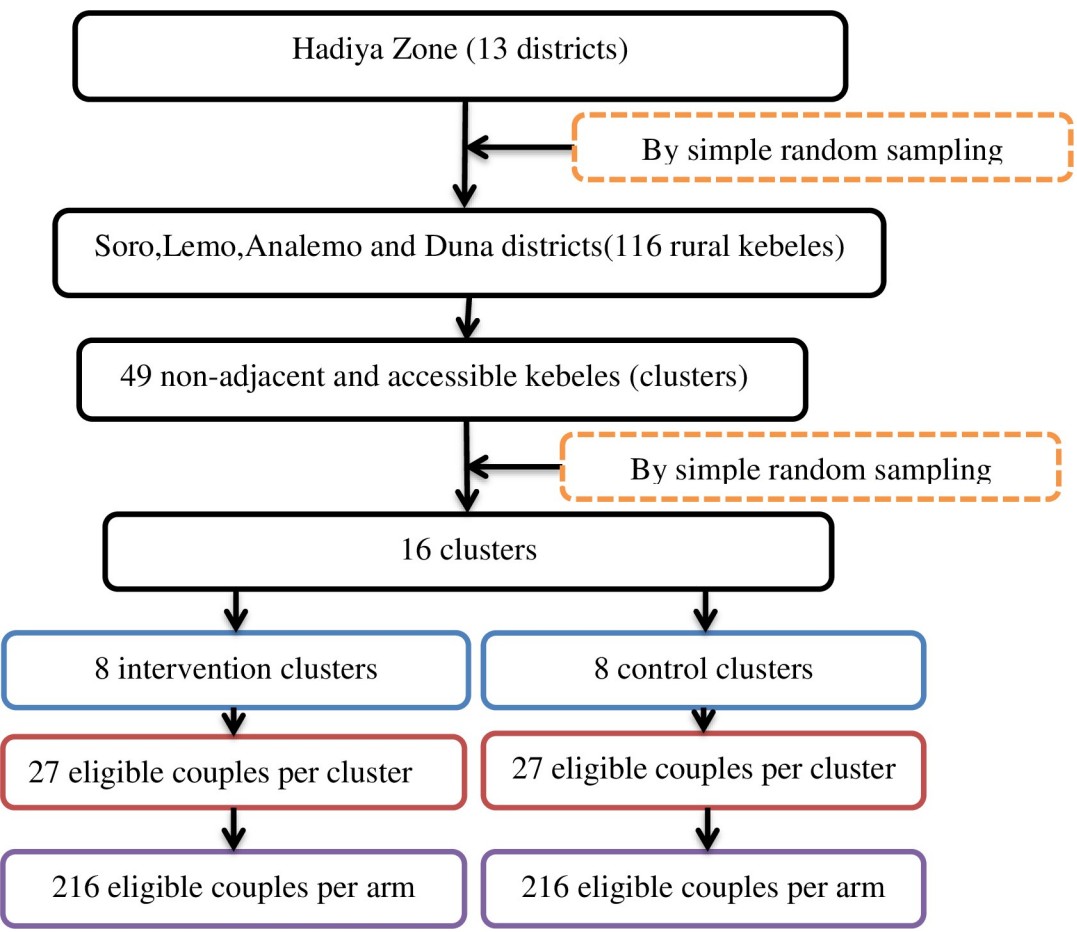

**Fig 2. Sampling procedure of the trial participants.**

four districts—will be selected by simple random sampling technique for the intervention and control groups. In each cluster, there will be 27 pregnant women with their husbands who will participate, making 216 couples per arm (Fig 2).

## Randomization

Kebeles in districts were used as the units of randomization for the trial. Randomizations of the clusters were carried out before recruiting the participants. From each of the four districts, four non-adjacent kebeles were selected. We stratified clusters into four study districts to evenly distribute known and unknown confounders into the intervention and control arms. A separate list of the clusters by study district was created, and unique codes were assigned to each cluster. The statistician, who was unaware of the study groups, divided four clusters in the stratum into blocks of size 2 based on the order in which clusters appeared alphabetically. By using sealed lots, the statistician selected the randomization sequence of clusters for each block from the two possible permutations within each block. Then, clusters in each block were assigned to intervention and control arms based on the selected permutation within the block for the stratum. The same procedures were applied to the remaining stratum clusters. Ultimately, two clusters were selected from each district, resulting in 8 clusters for the intervention arm and 8 clusters for the control arm with a 1:1 allocation ratio.

## Recruitment

We will review the Health Extension Workers logbook to identify pregnant women who are in their 1st trimester after randomization of clusters. Along with a logbook review, a pregnancy screening questionnaire that was adapted from Nega et al. [47] will be used not to miss pregnant women. If at least one of the six pregnancy screening questions is answered "yes," pregnancy will be ruled out. If a woman's response hinted at the possibility of pregnancy, she will be linked to a nearby healthcare facility (a hospital, health center, or clinic) for a pregnancy test (Table 1). All the randomization of the clusters into two groups will be done before identifying the couples in the trial. The identified couples will be invited to a meeting in a health post located in each chosen kebele, where data collectors will describe the study's purpose and nature in accordance with the information sheet. Before including participants in the trial, informed written and verbal consent will be acquired from them. According to their literacy level, couples who have been agreed to participate in the trial either sign or put their fingerprints. From June to July 2023, all pregnant women who fulfilled the eligibility criteria, gave consent to participate in the trial and live with their husbands in the chosen clusters were contacted and enrolled.

## Training for data collectors, supervisors, and health extension workers

Ten data collectors who are female diploma-holder nurses and two female supervisors with master's degrees in public health will be trained by the principal investigator for three days on the nature and purpose of the trial, the roles and responsibilities, research ethics during the survey, and interviewing skills. The role plays and demonstration of the interviewing skills will be carried out among the enumerators. How to use a mobile-based application, KoBo collect, will be an integral part of the training.

Health extension workers will be trained for three days. The trainers and participants manuals were adapted and modified from a psychosocial intervention to reduce gender-based violence in Kenya [48], effective prevention of IPV through couples training in Rwanda by Dunkle et al.[40] and effectiveness of a culturally appropriate intervention to IPV and HIV transmission in Ethiopia by Sharma et al. [39]. The training covers all the CBVPE contents, objectives, and delivery strategy in the trainer's manual. The trainer's manual is translated into the local language (Hadiyyisa) by experts in the field. The following combinations of teaching methods will be used during the training: brain storming, lectures, question and answer, take home-exercise and reflections, and role-plays.

## The intervention description

**Control group (study group 1):** the usual or routine care provided by HEWs will be considered a comparator for CBVPE. The couples in the control group will receive the standard

Table 1. Pregnancy screening questions adapted from Nega et al., Kersa, Ethiopia, 2013 [47].

| S. No | Items | Yes | No |
|---|---|---|---|
| 1 | During the last four weeks, did you give birth? | | |
| 2 | Do you fully breastfeed or are you less than six months postpartum and have not experienced menstrual bleeding since bearing your child? | | |
| 3 | During the last week, did your last period begin? | | |
| 4 | In the last week, did you experience a miscarriage or an abortion? | | |
| 5 | Since your last period, have you avoided sexual intercourse? | | |
| 6 | Have you been consistently and appropriately using a safe method of contraception (pills, injectables, and Norplant)? | | |

health extension package delivered by HEWs during pregnancy and postpartum. Usual standard care provided by Health extension workers are preventive and promotive services which are categorized under four main topic areas: family health services, disease prevention and control, hygiene and environmental sanitation, health education and communication.

## Intervention group

Pregnant women and their husbands will receive CBVPE from the $2^{nd}$ trimester of pregnancy to delivery for six months. The intervention package is aimed at raising knowledge and awareness of Gender-Based Violence (GBV), Violence Against Women (VAW) and IPV, the consequences and common triggers of IPV during pregnancy, inequitable gender norms and their consequences, unhealthy and healthy relationships, power and control in relationships, joint decision-making, communication, and conflict resolution. After adequate training, HEWs will provide health education to couples in the intervention arms at health-post in group. It will be provided at every month for six consecutive times. Each session may take about 60 minutes. All sessions will contain interactive lectures, take-home exercises, and reflection. During class sessions, flip chart paper and posters will be used. Posters with session key messages will be posted in each session and distributed to participants at the end. Each session is designed with the intention that they will be carried out in an enclosed workshop space. Neither the HEWs nor the participants will be informed about the intervention hypothesis.

## Intervention fidelity

Intervention fidelity refers to the degree to which components of intervention are executed as designed [49]. To enhance the intervention's fidelity, we will implement the following actions: a trainer manual has already been developed; extensive training will be provided for interventionists (HEWs); spot checks will be conducted by field supervisors during each intervention session; and feedback meetings with interventionists will be held after each session. A fidelity checklist that addresses adherence and intensity of the intervention will be used to evaluate intervention fidelity at the end of each session.

## Appointment and follow-up for intervention group

The contact address of each couple included in the study will be taken. The addresses of the couples, including kebele, gote (sub-kebele), house number (if applicable), and telephone number, will be taken by the researcher and the HEWs (educators). At the end of each session, the educator and the participants will agree on the date of the next visit, which will be recorded by the educator in her diary and on a similar diary that will be left with the participants. Before two days of the next visit, the educator reminds the couples about the next session by calling or messaging them. Either of the couples who missed the classroom sessions will be visited by HEWs at their home, trying to understand the reason why they missed an appointment. Then another classroom session will be arranged. At the end of each session, the number of couples who attended will be recorded. The authors will also identify which couple attended each session. Couples in the intervention clusters will be visited three times in case of absence to improve the participants' compliance. The end-line, 8-month assessment will be conducted after the intervention is complete. Every month, HEWs will screen the couples for any dispute or disagreement due to intervention during a home visit by asking either of the couples. If so, she negotiates with the couples. If the dispute still continues in the second screening, she communicates with the principal investigator, and they both resolve the dispute between the couples as much as possible.

## Outcome assessment

**Primary outcome.** The primary outcome of the trial is IPV during pregnancy among women (experience IPV versus do not experience IPV).

**Intimate partner violence** will be measured as the percentage of pregnant women who experience physical, psychological, or sexual abuse from their husbands during pregnancy [2].

**Physical intimate partner violence**: a pregnant woman will be asked six physical intimate partner violence questions about whether her husband (i) slapped or threw something at her that could hurt her, (ii) pushed, shoved, or pulled her hair, (iii) "hit her with his fist or something else that could harm her", (iv) kicked, dragged, or beat her up", (v) "tried to choke her or burn her on purpose" and (vi) "threaten to use or actually use a gun, knife, or other weapon against her". If a pregnant mother responds to one or more of these questions about physical intimate partner violence, it will be recorded as 1 = yes (experienced physical intimate partner violence); otherwise, 0 = no (not experienced physical intimate partner violence) [2].

**Sexual intimate partner violence**: a pregnant woman will be asked three sexual intimate partner violence questions: whether her husband (i) physically forced her to have sexual intercourse with him when she did not want to", (ii) "forced her with threats or in any other way to perform sexual acts she did not want to," and (iii) physically forced her to perform any other sexual acts she did not want to" during the most recent pregnancy. If the pregnant mother responds to one or more of these sexual intimate partner violence questions, it will be recorded as 1 = yes (experienced sexual IPV); otherwise, 0 = no (not experienced any sexual IPV) [2,24].

**Psychological or emotional intimate partner violence** will be measured by asking the pregnant mother the following three psychological violence questions: whether her husband (i) said or did something to humiliate her in front of another person; (ii) threatened to hurt or harm her or someone she cared about; and (iii) insulted or made her feel bad about herself. If the mother responds to one of these emotional or psychological IPV questions, it will be recorded as 1 = yes; otherwise, 0 = no (she has not experienced any emotional or psychological intimate partner violence during this pregnancy) [2,24].

**Secondary outcomes.** Secondary outcomes are couples' knowledge, attitudes, and husbands' controlling behavior toward IPV and women's autonomy and self-efficacy.

**Knowledge of IPV:** To measure knowledge, couples will be asked about nine items of IPV knowledge questions. Couples scoring approximately 50% and above of the correct answer will be scored as "good knowledge," and less than 50% will be scored as "poor knowledge" [25].

**Attitude towards IPV**: Couples will be asked whether or not a husband is justified in hitting or beating his spouse in 5 scenarios: (a) "If she is going out without telling him? (b) "If the wife neglects the children? (c) "If the wife argues with husband? (d) "If wife refuses to have sex with husband? (e) "If wife burns the food? Responses of "supportive attitude toward IPV" to one or more of the scenarios will be coded 1; "not supportive attitude toward IPV" to all scenarios will be coded "0" [50].

**Controlling behavior**: couples will be asked, Is the husband justified in controlling the wife in the following situation? Whether: (i) "jealous if she talked with other men", (ii) "accused her of unfaithfulness", (iii) "do not permit her to meet her friends, (iv) "tried to limit her contact with family, (v) "insisted on knowing where she is, and "did not trust her with money". Couples responded "yes" to one or more of the control questions coded as 1. Otherwise, it will be coded 'no' as 0 [27].

**Women's autonomy**: Women's decision-making autonomy will be assessed by asking couples who decide on the following five issues: household purchases, including small and large ones; visiting relatives and friends; spending the wife's earnings; the number of children to have; and obtaining health care for her. The response options will be "husband only", wife

only, both husband and wife or "other person or family member". Women will be considered to participate in decision-making if they make decisions alone or jointly with their husbands. Women will be scored one (1) for answers to each variable that included her (alone or jointly) in decision-making; otherwise, they will be scored zero (0). The index of household decision-making power (autonomy) contains five variables, so the respondents are scored from 0 to 5. Two binary measures from these indexes will be created to indicate women with higher autonomy (score of ≥3) versus lower autonomy (score ≤ 2) [51].

**Women's self-efficacy** is the capacity perceived by an individual to perform a successful behavior. Questionnaire assessing women's self-efficacy was adapted and modified from General self-efficacy scale developed by Peng et al. [52]. It consists of 10 items scored on a 4-point Likert scale, from 1 (not at all true) to 4 (exactly true). The total scale ranges from 10 to 40, with higher self-efficacy (a score of ≥ 20) and lower self-efficacy (a score of < 20) [52].

## Data collection methods

Structured questionnaires will be used to collect data that is developed and adapted from a WHO multicountry study on intimate partner violence [2]. The questionnaire will be pretested for clarity, logical flow, cultural appropriateness, etc. and amendments will be made as needed. As WHO recommends the female data collectors to enhance disclosure [53], ten data collectors who are diploma-holder nurses will be recruited. They will be trained on questionnaire contents and interviewing techniques. A structured questionnaire prepared in Amharic and the local language (Hadiyisa) will be used to collect data. Data will be collected at baseline and end-line through face-to-face interviews using a mobile application, KoBo Collect. At baseline, data will be collected on sociodemographic, socioeconomic, and maternal factors, whereas, at baseline and end-line, data will be collected on knowledge, attitude, controlling behavior, women's autonomy and self-efficacy, and experience of IPV. Data will be collected separately from the wife and husband. We will assign a different team to carry out the intervention and collect the data for the purpose of masking however; the same data collectors will be deployed baseline and endline.

## Instrument translation, validity and reliability

An essential step for the questionnaire's validity is the translation of the original English version into a native language [54]. We'll follow a methodical procedure suggested by Beaton et al. [55] to maintain conceptual, experiential, and idiomatic equivalence. Bilingual translators will translate the instruments into Hadiyisa. The instruments will be translated back into English by two bilingual translators from the Department of Hadiyisa, Wachemo University, who are completely blind to the original English version. To establish semantic equivalence, the back-translation is necessary. To combine all language versions and create the pre-final questionnaire for field testing, an expert committee meeting will be organized [56]. To verify the translation's accuracy, the original and back-translated English versions of the instruments will be compared. Using a technique developed by Skperber et al. [57], each item in the original and back-translated versions will be ranked in terms of comparability of language and similarity of interpretability for the purpose of certifying the translated instrument. Fluent in English experts from the department of foreign languages and literatures will assess the comparisons using Likert scales from 1 (highly comparable/extremely similar) to 7 (not at all comparable/not at all similar). There will be a formal evaluation of the translation for any mean score greater than 3 S2 File.

Pilot testing of the translated tool will be conducted among 27 couples who will not be included in the actual trial to assess its understanding, comprehension, and appropriateness.

Additionally, subject experts will evaluate the content, and the necessary changes will be made. The tools' internal consistency will be tested using Cronbach's alpha. A tool with a Crombach alpha of $\geq 0.7$ will be considered reliable [58]. Finally, the questionnaires will be utilized at the study's baseline and completion.

## Data management

Regular supervision and follow-up will be done by supervisors and the principal investigator. A regular check-up of the data will be made on a daily basis by supervisors. Every day, data collected by KoBo Collect (the mobile application) will be checked for completeness and consistency before being sent to the server (KoBo Toolbox). The supervisor and principal investigator will consider the following points during the checkup of the collected data: the length of time to finish each survey, the collection of Global Positioning System (GPS) points, and their random distribution. Trial participants who are loss to follow up will be listed along with their reasons. Only the study team will have access to the data set. The anonymous record and coding of personal data will not be disclosed to any third party to ensure strict secrecy. Couple code will be used to link data collected from wife and husband. Data will be stored on a password-protected data server (KoBo Toolbox) and only accessible by the research team.

## Data processing and analysis

A complete questionnaire with individual code and specific cluster will be exported from server to SPSS or Stata for analysis. At baseline, frequency, percentages, means, and standard deviations will be used to describe the study population in the intervention and control arms before starting the intervention. The results will be displayed in figures, diagrams and tables. The baseline characteristics of the participants and their prior exposure to IPV will be examined and compared between the intervention and control groups. The chi-square test and independent sample t-test will be used to compare the baseline characteristics for categorical and continuous variables, respectively. After the end-line survey, the proportion of women who experienced IPV during the trial period will be calculated with a corresponding confidence interval.

We will use the McNemar test to assess the differences in outcomes of interest in both intervention and control groups before and after the intervention for categorical data. A paired t-test will be used to compare continuous outcome of interest in the intervention and the control groups after and before the intervention. A model of longitudinal data analysis, the GEE (Generalized Estimating Equation), will be used to test the independent effect of the intervention on the outcome of interest. A working correlation structure will be specified to account within cluster correlation. The choice of preferred working correlation structure will be based on the correlation structure matrix of the observed data. Data analysis will be performed with an intention-to-treat analysis approach. During the analysis, the effect size, confidence interval, and p-value will be calculated. The Quasi-likelihood under the Independence Model Criterion (QIC) and the Akaike Information Criterion (AIC) will be used to assess model fitness. The QIC and AIC values will be calculated for each model and compared. A model with lower diagnostic criteria (deviance or -2 x log-likelihood ratio) will be used as the best-fitted model. Sensitivity analysis will be conducted to assess the robustness of the intervention effect by changing the assumptions made in the model, such as the working correlation structure, outliers, missing data, and controlling potential confounders.

## Ethical consideration

The study protocol was approved by Ethics committee of Jimma University's Institutional Review Board on November 8, 2022 (JUIH/IRB/222/2022) S1 File.

**Table 2. Participants' timeline of enrolment, implementation and assessment schedule.**

| Activity | Study period | | | | | | | | | | |
|---|---|---|---|---|---|---|---|---|---|---|---|
| | Enrolment | Allocation | Intervention | | | | | | | | Close-out |
| | Month 1 | Month 2 | Months | | | | | | | | Month 11 |
| | | | 3 | 4 | 5 | 6 | 7 | 8 | 9 | | |
| Enrolment | | | | | | | | | | | |
| Eligibility screen | ■ | | | | | | | | | | |
| Informed consent | ■ | | | | | | | | | | |
| Allocation | | ■ | | | | | | | | | |
| Baseline assessments | | ■ | | | | | | | | | |
| Interventions | | | ■ | ■ | ■ | ■ | ■ | ■ | ■ | | |
| Endline assessments | | | | | | | | | | | ■ |

## Timeline of the study

Both primary and secondary outcomes will be assessed as illustrated participants' timeline of enrolment, implementation and assessment schedule (Table 2) and S1 Fig [59].

## Trial status

Both educators' and participants' manuals were prepared in both Amharic (the national language) and Hadiyisa (the local language). Training of data collectors and HEWs (intervention implementers) was completed. All the study participants in the intervention and control arms were recruited. Baseline data: sociodemographic and socioeconomic characteristics, reproductive history, knowledge, attitude and controlling behaviour toward IPV, women's autonomy and self-efficacy and IPV in last 12 months and recent pregnancy were collected while the first session of the intervention is being implemented.

## Data monitoring

Monitoring is an important aspect of improving the quality of data. Field supervisors will be in charge of monitoring and auditing the data during the baseline, implementation, and endline periods.

## Discussion

Ethiopia is one of the countries striving to achieve SDG 5.2 calling for the elimination of all forms of violence against women and girls. Particularly in an Ethiopian context, interventions involving spouses are not well-researched. Locally adapted intervention strategies are important to control or reduce violence during pregnancy [46]. If couple-based violence prevention education proves to be effective in reducing intimate partner violence during pregnancy in the study setting, it will be scaled up in other parts of Ethiopia. The trial's protocol will be made public in a peer-reviewed, open-access journal to increase transparency. The trial registry S2 Table and Jimma University's research ethics committee will be informed of any changes to any aspect of the study. The results will be presented at conferences and workshops held on a national and worldwide scale. The trial findings will appear in open-access, peer-reviewed journals. The study protocol was developed based on the guidance of the SPIRIT (Standard Protocol Items: Recommendations for Interventional Trials) checklist S1 Table.

## Supporting information

**S1 Fig. SPIRIT schedule of enrollment, interventions and assessments.**
(DOC)

**S1 Table. SPIRIT 2013 checklist.**
(DOC)

**S2 Table. Items from the WHO trial registration data set.**
(DOCX)

**S1 File. Institutional Review Board (IRB) approval.**
(PDF)

**S2 File. Study protocol.**
(DOCX)

## Acknowledgments

The study team wishes to acknowledge the support delivered by Jimma University and Wachemo Uniiversity.

## Author Contributions

**Conceptualization:** Zeleke Dutamo Agde, Nega Assefa, Muluemebet Abera Wordofa.

**Methodology:** Zeleke Dutamo Agde, Jeanette H. Magnus, Nega Assefa, Muluemebet Abera Wordofa.

**Writing – original draft:** Zeleke Dutamo Agde.

**Writing – review & editing:** Jeanette H. Magnus, Nega Assefa, Muluemebet Abera Wordofa.

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
