## [Decision Letter · Decision Letter 0]

15 Dec 2023

PONE-D-23-32195

The effect of couple-based violence prevention education on intimate partner violence during pregnancy in Southwest Ethiopia: study protocol for a cluster randomized controlled trial

PLOS ONE

Dear Dr. Agde,

Thank you for submitting your manuscript to PLOS ONE. After careful consideration, we feel that it has merit but does not fully meet PLOS ONE’s publication criteria as it currently stands. Therefore, we invite you to submit a revised version of the manuscript that addresses the points raised during the review process.

We look forward to receiving your revised manuscript.

Kind regards,

Azmeraw Ambachew Kebede, MSc

Academic Editor

PLOS ONE

*Comments from PLOS Editorial Office: We note that one or more reviewers has provided references to specific previously published works. As always, we recommend that you please review and evaluate the requested works to determine whether they are relevant and should be cited. It is not a requirement to cite these works. We appreciate your attention to this request.*

Reviewers' comments:

Reviewer's Responses to Questions

**Comments to the Author**

1. Does the manuscript provide a valid rationale for the proposed study, with clearly identified and justified research questions?

Reviewer #1: No

Reviewer #2: Yes

2. Is the protocol technically sound and planned in a manner that will lead to a meaningful outcome and allow testing the stated hypotheses?

Reviewer #1: Yes

Reviewer #2: Partly

3. Is the methodology feasible and described in sufficient detail to allow the work to be replicable?

Reviewer #1: Yes

Reviewer #2: No

4. Have the authors described where all data underlying the findings will be made available when the study is complete?

Reviewer #1: Yes

Reviewer #2: Yes

5. Is the manuscript presented in an intelligible fashion and written in standard English?

Reviewer #1: Yes

Reviewer #2: No

6. Review Comments to the Author

You may also provide optional suggestions and comments to authors that they might find helpful in planning their study.

Reviewer #1: See attached review. See attached review. See attached review. See attached review. See attached review.

Reviewer #2: The authors present the protocol for a cluster randomized controlled trial to evaluate couple-based violence prevention education and its ability to reduce intimate partner violence during pregnancy in Southwest Ethiopia. Authors provide details on the rationale, setting, design and methods, though there are some aspects that require clarification or additional information. The manuscript will be strengthened if the authors consider the following points.

1. In the sample size determination section, it is not clear what was calculated versus what was assumed. The way it is presented, all of the information was set. Authors need to clarify what they assumed and what was actually calculated. Also, a cluster size of 27 for each of 8 clusters in a group would result in 216 couples, not 213 couples, so that should also be clarified.

2. line 169: "applying permuted block randomization methods" is too broad. For a protocol paper, authors should provide exactly what was (or will be) used to do the randomization.

3. Is the "Appointment and follow-up section just for the intervention group? This should be clarified. Also, since the intervention is administered in a group format, the entire group agrees on the date of the next session? (I ask, because line 231 says the "educator and the participant" which suggests individual appointments rather than a group session.

4. line 236 - will authors also identify which couples attend each session (in addition to the number of couples who attended). That might be useful should authors decide to do any follow-up analyses that look at extent of exposure to the intervention on the outcomes.

5. Authors should provide more information about the planned GEE models. How is cluster being accounted for? Will authors use the baseline outcome as a covariate in a model for the 8 month assessment or do they plan to model the two timepoints for each couple? It looks like some information will be collected separately from the wife and from the husband. How will that be handled? Are there any covariates to be included in the models? Are there any planned sub-group analyses or sensitivity analyses? Authors mention using QIC to select the best-fitted models - which models are being compared (this doesn't need to be the exact models, but authors should specify what aspects of the models they are using QIC to help identify).

Minor edits

1. lines 31 and 361: "STATA" should be "Stata" (https://www.statalist.org/forums/help#spelling)

2. lines 32 and 370: "appreciate" should be "assess"

3. lines 34 and 372: "continues" should be "continuous"

4. line 66: "such antepartum" should be "such as antepartum"

5. lines 70 and 72: authors have not defined SDG or IPV

6. Authors define CBVPE multiple times (for example, lines 85, 88, 92, 95) - it only needs to be defined once in the body of the manuscript and then can be used throughout the rest of the manuscript).

7. line 116: "which is one of the ..." is confusing since the start of the sentence talked about 4 out of the 13 districts...authors should clarify.

8. lines 138-139: "Couples who lived at least six months.." (lived together?, lived in the study area?) - this should be clarified.

9. line 166 - authors mention an alphabetical list - alphabetical list of what?

10. line 173: "trimester before." - before what? I think a word or words is missing.

11. lines 178-179 contradicts what is said in lines 164-165

12. lines 184-185: authors describe something that took place in June-July 2023, so they should use past tense not future tense ("were" instead of "will be"). Also, not all pregnant women living with their husbands in the clusters were enrolled, right? They had to meet eligibility criteria and consent...

13. There are numerous places in the manuscript where lines are duplicated (for example, lines 192-194, 277-279, 281-282m 284-287). Authors should carefully read the manuscript to ensure information is not repeated.

14. line 211 - authors mention 4 main topics, but it looks like there are 5 topics listed (unless health education and communication is considered one topic).

15. line 235: "rule out the reason for the missing" should be rephrased - I don't think authors are ruling out anything, maybe trying to understand the reason for why a couple missed an appointment.

16. lines 238-239: "after two months of the intervention" should be rephrased, since I believe the 8 month assessment is being done after the intervention is complete.

17. line 240-241 - would a disagreement or dispute be due to the intervention?

18. line 246: authors are not capturing the magnitude of intimate partner violence - they are just capturing whether or not it happened.

19. line 362: "tables, diagrams, and figures" are not descriptive statistics - they are ways in which to display descriptive statistics. Authors should clarify their statement.

7. PLOS authors have the option to publish the peer review history of their article (what does this mean?). If published, this will include your full peer review and any attached files.

Reviewer #1: No

Reviewer #2: No

---

## [Author Response · Author response to Decision Letter 0]

18 Jan 2024

We have addressed in the rebuttal letter.

---

## [Decision Letter · Decision Letter 1]

11 Apr 2024

PONE-D-23-32195R1The protocol for a cluster randomized controlled trial to evaluate couple-based violence prevention education and its ability to reduce intimate partner violence during pregnancy in Southwest Ethiopia.PLOS ONE

Dear Dr. Agde,

Thank you for submitting your manuscript to PLOS ONE. After careful consideration, we feel that it has merit but does not fully meet PLOS ONE’s publication criteria as it currently stands. Therefore, we invite you to submit a revised version of the manuscript that addresses the points raised during the review process.

Thank you for your diligent work addressing all of the reviewer's comments. There are two minor edits that need to be addressed prior to publication. Once we receive the revised manuscript, it will not need to go out to reviewer's again.

We look forward to receiving your revised manuscript.

Kind regards,

Michelle L. Munro-Kramer, PhD, CNM, FNP-BC, FAAN

Academic Editor

PLOS ONE

Journal Requirements:

Additional Editor Comments:

 The following changes are requested:

1) Per PLOS One guidelines - please use three heading levels that are all the same size font and bolded.

2) When stating author's names in the text, please use the last name only. For example,

a) Pg 5 of the revised manuscript should be Leight et al. (not Jessica)

b) Pg 11 of the revised manuscript should be Nega et al.

c) Pg 12 of the revised manuscript should be Dunkle et al.

d) Pg 12 of the revised manuscript should be Sharma et al.

e) Pg 18 of the revised manuscript should be Peng et al.

Reviewers' comments:

Reviewer's Responses to Questions

**Comments to the Author**

1. Does the manuscript provide a valid rationale for the proposed study, with clearly identified and justified research questions?

Reviewer #1: Yes

Reviewer #2: Yes

2. Is the protocol technically sound and planned in a manner that will lead to a meaningful outcome and allow testing the stated hypotheses?

Reviewer #1: Partly

Reviewer #2: Yes

3. Is the methodology feasible and described in sufficient detail to allow the work to be replicable?

Reviewer #1: Yes

Reviewer #2: Yes

4. Have the authors described where all data underlying the findings will be made available when the study is complete?

Reviewer #1: Yes

Reviewer #2: No

5. Is the manuscript presented in an intelligible fashion and written in standard English?

Reviewer #1: Yes

Reviewer #2: Yes

6. Review Comments to the Author

You may also provide optional suggestions and comments to authors that they might find helpful in planning their study.

Reviewer #1: Thank you for the revisions to the protocol. It is now ready to be published and will be a useful contribution.

Reviewer #2: The authors have addressed all of my earlier concerns. I do not have any further concerns to raise at this time.

7. PLOS authors have the option to publish the peer review history of their article (what does this mean?). If published, this will include your full peer review and any attached files.

Reviewer #1: No

Reviewer #2: No

---

## [Author Response · Author response to Decision Letter 1]

13 Apr 2024

Addressed in the revised manuscript and rebuttal letter

---

## [Editor Report · Decision Letter 2]

19 Apr 2024

The protocol for a cluster randomized controlled trial to evaluate couple-based violence prevention education and its ability to reduce intimate partner violence during pregnancy in Southwest Ethiopia.

PONE-D-23-32195R2

Dear Dr. Agde,

We’re pleased to inform you that your manuscript has been judged scientifically suitable for publication and will be formally accepted for publication once it meets all outstanding technical requirements.

Kind regards,

Michelle L. Munro-Kramer, PhD, CNM, FNP-BC, FAAN

Academic Editor

PLOS ONE

Additional Editor Comments (optional):

Thank you for addressing the minor editorial comments so quickly. We are pleased to accept your manuscript!
---

## [Editor Report · Acceptance letter]

30 Apr 2024

PONE-D-23-32195R2 

PLOS ONE

Dear Dr. Agde, 

I'm pleased to inform you that your manuscript has been deemed suitable for publication in PLOS ONE. Congratulations! Your manuscript is now being handed over to our production team.

Kind regards, 

on behalf of

Dr. Michelle L. Munro-Kramer 

Academic Editor

PLOS ONE